# Comparison between the Regenerative and Therapeutic Impacts of Bone Marrow Mesenchymal Stem Cells and Adipose Mesenchymal Stem Cells Pre-Treated with Melatonin on Liver Fibrosis

**DOI:** 10.3390/biom14030297

**Published:** 2024-03-01

**Authors:** Ahmed Elzainy, Abir El Sadik, Waleed Mohammad Altowayan

**Affiliations:** 1Department of Anatomy and Histology, College of Medicine, Qassim University, Buraydah 51452, Saudi Arabia; ahmedelzainy@qu.edu.sa (A.E.); abeer.ouaida@kasralainy.edu.eg (A.E.S.); 2Department of Anatomy and Embryology, College of Medicine, Cairo University, Cairo 11956, Egypt; 3Department of Pharmacy Practice, College of Pharmacy, Qassim University, Buraydah 51452, Saudi Arabia

**Keywords:** proliferation, inflammation, apoptosis, regeneration, therapy, liver

## Abstract

Background: The distinctive feature of liver fibrosis is the progressive replacement of healthy hepatic cells by the extracellular matrix protein, which is abundant in collagen I and III, with impaired matrix remodeling. The activation of myofibroblastic cells enhances the fibrogenic response of complex interactions of hepatic stellate cells, fibroblasts, and inflammatory cells to produce the excessive deposition of the extracellular protein matrix. This process is activated by multiple fibrogenic mediators and cytokines, such as TNF-α and IL-1β, accompanied with a decrease in the anti-fibrogenic factor NF-κβ. Mesenchymal stem cells (MSCs) represent a promising therapy for liver fibrosis, allowing for a more advanced regenerative influence when cultured with extrinsic or intrinsic proliferative factors, cytokines, antioxidants, growth factors, and hormones such as melatonin (MT). However, previous studies showed conflicting findings concerning the therapeutic effects of adipose (AD) and bone marrow (BM) MSCs; therefore, the present work aimed to conduct a comparative and comprehensive study investigating the impact of MT pre-treatment on the immunomodulatory, anti-inflammatory, and anti-apoptotic effects of AD- and BM-MSCs and to critically analyze whether MT-pre-treated AD-MSCs and BM-MSCs reveal equal or different therapeutic and regenerative potentials in a CCl4-injured liver experimental rat model. Materials and methods: Six groups of experimental rats were used, with ten rats in each group: group I (control group), group II (CCl4-treated group), group III (CCl4- and BM-MSC-treated group), group IV (CCl4 and MT-pre-treated BM-MSC group), group V (CCl4- and AD-MSC-treated group), and group VI (CCl4 and MT-pre-treated AD-MSC group). Liver function tests and the gene expression of inflammatory, fibrogenic, apoptotic, and proliferative factors were analyzed. Histological and immunohistochemical changes were assessed. Results: The present study compared the ability of AD- and BM-MSCs, with and without MT pre-treatment, to reduce hepatic fibrosis. Both types of MSCs improved hepatocyte function by reducing the serum levels of ALT, aspartate aminotransferase (AST), alkaline phosphatase (AKP), and total bilirubin (TBIL). In addition, the changes in the hepatocellular architecture, including the hepatocytes, liver sinusoids, central veins, portal veins, biliary ducts, and hepatic arteries, showed a decrease in hepatocyte injury and cholestasis with a reduction in inflammation, apoptosis, and necrosis of the hepatic cells, together with an inhibition of liver tissue fibrosis. These results were augmented by an analysis of the expression of the pro-inflammatory cytokines TNFα and IL-1β, the anti-fibrogenic factor NF-κβ, the apoptotic factor caspase-3, and the proliferative indicators antigen Ki-67 and proliferating cell nuclear antigen (PCNA). These findings were found to be statistically significant, with the restoration of normal parameters in the rats that received AD-MSCs pre-treated with MT, denoting optimal regenerative and therapeutic effects. Conclusions: AD-MSCs pre-treated with MT are the preferred choice in improving hepatic fibrosis and promoting the therapeutic and regenerative ability of liver tissue. They represent a very significant tool for future stem cell use in the tissue regeneration strategy for the treatment of liver diseases.

## 1. Introduction

Liver cirrhosis, the progression of liver fibrosis, is currently the eleventh most common cause of mortality worldwide, and its prevalence is continuously increasing [1,2,3]. Liver fibrosis is considered one of the most serious health problems, producing changes in the architecture and vasculature of some organs and interrupting normal function [4,5]. Liver fibrosis is a wound-healing procedure in which multiple consequences are induced, including immune cell infiltration, myofibroblast activation, hepatocyte apoptosis, and necrosis [3]. Liver fibrosis features the gradual replacement of functional hepatic cells by the extracellular matrix protein, rich in collagen I and III, with impaired matrix remodeling [6,7,8,9]. The excessive deposition of the extracellular matrix is triggered by a fibrogenic response of complex interactions of matrix-forming hepatic stellate cells, fibroblasts, and inflammatory cells. Hepatic stellate cells are allocated between the hepatocytes and sinusoid endothelial cells. They are activated by fibrogenic mediators and cytokines to differentiate into active myofibroblasts with the expression of pro-inflammatory mediators such as alpha-smooth muscle actin (α-SMA), tumor necrosis factor-alpha (TNF-α), and interleukin 1 beta (IL-1β) [10,11,12,13,14]. The transforming growth factor-beta (TGF-β) signaling pathway is a main regulator inducing liver fibrosis. It stimulates the formation of heteromeric complexes and the phosphorylation of particular proteins, such as the receptor-regulated ®-Smads, Smad2, and Smad3, to initiate intracellular signaling. These transcription factors control the profibrotic genes [15]. Another intracellular signaling pathway that is involved in the pathophysiology of liver fibrosis is the Wnt pathway, which is β-catenin-dependant. β-catenin is a multiprotein complex, with Axin, GSK-3 β, β -TrCP, APC, and CK1α, and acts as an adhesion molecule and a transcription factor. In the injured liver, it is transferred from the cell membrane into the cell cytoplasm. The active Wnt pathway stimulates the dissociation of this multiprotein complex, the inactivation of GSK-3 β, and the regulation of inflammatory cytokines, leading to the activation of hepatocyte stellate cells [16]. Numerous signaling pathways connected to inflammation, proliferation, and apoptosis can be triggered by TNF-α. TNFα has been associated with the pathogenesis of chronic liver inflammation with the activation of local hepatic stellate cells into fibrogenic myofibroblasts, leading to liver fibrosis [10]. Consequently, the control of hepatic stellate cell activation could potentially elicit the anti-fibrotic process [17]. Therefore, extracellular matrix degradation and remodeling, by blocking myofibroblast signaling, are crucial in reversing liver fibrosis and limiting its progression to cirrhosis. One significant and early mediator of fibrosis is IL-1β. This is induced within one hour of the profibrotic procedure. It mediates the extracellular matrix degradation, causing the hepatic sinusoids to collapse by inducing matrix metalloproteinase-9 (MMP-9) expression [18]. Furthermore, IL-1β boosts activated hepatic stellate cells’ ability to survive [19]. The transcription and anti-fibrogenic element nuclear factor kappa β (NF-κβ) is the principal regulator of inflammation and cell necrosis. Its normal activity in hepatocytes is to protect against liver fibrosis by inhibiting hepatocyte death. At the same time, decreased levels of NF-κβ increase hepatocellular injury and fibrosis. NF-κβ principally moderates hepatic fibrogenesis by controlling three different mechanisms. It regulates the first trigger of fibrogenic response, which is hepatocyte injury. It reduces the inflammatory signals produced in macrophages and other inflammatory cells, and it controls the fibrogenic responses in hepatocyte stellate cells [20]. Therefore, regulation of the pro-inflammatory moderators TNFα and IL-1β and an increase in the expression of anti-fibrogenic factor NF-κβ is crucial for hepatocyte therapy and the regeneration of injured hepatocytes.

Mesenchymal stem cells (MSCs) represent a potential therapeutic strategy for liver fibrosis [21,22,23,24]. They are the most broadly applied type of stem cell and can be obtained from multiple sources, such as bone marrow, adipose tissue, umbilical cord, liver tissue, menstrual blood, and the placenta [25,26,27]. The regenerative potential of MSCs was attributed to their highly proliferative, multipotent, and differentiation capacities allowing them to replace damaged cells. However, it is currently believed that the paracrine action of MSCs is responsible for the production of some bioactive factors, stimulating the subject’s own progenitor cells’ maturation and proliferation. In addition, they stimulate the release of immunoregulatory mediators which modulate the cascade of inflammation [28,29,30]. Therefore, the therapeutic effects of MSCs in liver fibrosis are predisposed by the secretion of these factors rather than their ability to transform into hepatocytes which replace the damaged parenchymal cells [31]. The injured hepatocytes undergo apoptosis and necrosis influenced by multiple stimulators and signal pathways including transcription factors, inflammatory cytokines, chemokines, kinases, apoptotic mediators, growth factors, and oxidative stress products [32,33]. Nevertheless, some studies reported that the profibrotic properties of MSCs are attributed to these cells having a myofibroblastic phenotype in vitro or the mesenchymal origin of myofibroblasts revealing profibrotic potentials. Therefore, further investigations are needed to evaluate these effects as most of the studies were implemented in the early stages of fibrosis, although fibrosis is commonly diagnosed in more advanced stages [34]. Also, recent studies confirmed that the MSC treatment of a target organ could have a therapeutic role for a distant organ, as proved by Mouiseddine et al. [35]. The authors detected that MSC infusion indirectly improved liver functions through the prevention of gut injuries. Moreover, one of the challenges facing MSC therapy is the transplantation conditions such as the timing after injury. Several studies proposed that earlier administration improves the therapeutic effects of MSCs on fibrosis [36]. Another limitation of the use of MSCs is their source. For example, adipose tissue MSCs are easier to harvest, and they yield higher numbers of MSCs. In addition, the major concern in the usage of MSCs is the induction of certain pathological conditions, mainly tumor formation. Consequently, additional studies are needed to evaluate the optimal conditions for the administration of stem cells. Bone marrow (BM) MSCs could improve liver fibrosis in cirrhotic rat models [37]. The authors reported a significant decrease in the collagen proportionate area and the downregulation of the TGF-B1/Smad signaling pathway. In addition, liver function was improved, and fibrosis progression and hepatocyte necrosis were diminished. On the other hand, adipose tissue (AD) MSCs have been proposed for cell-based therapy as a substitute for BM-MSCs. As a source of multipotent stem cells, AD-MSCs can be more easily isolated. They proliferate into larger numbers of cells and continue to proliferate for 21 days more than the BM-MSCs. AD-MSCs were found to yield more stem cells than BM-MSCs and differentiate in vitro and in vivo into hepatocytes [38]. Moreover, BM-MSCs revealed more early senescence during expansion than AD-MSCs [39,40]. Both AD- and BM-MSCs are influenced by many extrinsic and intrinsic signaling factors such as cytokines, antioxidants, growth factors, extracellular matrix, enzymes, and hormones. An example of these hormones is melatonin (MT), which plays an important role in the regulation of cell proliferation. It has anti-inflammatory and antioxidant activities [24]. Recent studies determined that MT enhances the therapeutic effects of MSCs by promoting their self-renewal capability and homing potentials and reducing the inflammatory modulators’ expression [41,42]. Zhou et al. [43] detected that MT could prevent the replicative senescence of the AD-MSCs. The maintenance of the biological activities of MSCs is controlled by multiple complex mechanisms including intracellular and extracellular signaling pathways and the presence of potential immunomodulatory mediators in a favorable microenvironment. Therefore, the present work aimed to accomplish a comparative and comprehensive study investigating whether MT pre-treatment could improve the immunomodulatory, anti-inflammatory, and anti-apoptotic effects of AD- and BM-MSCs with an analysis of the expression of the anti-inflammatory cytokines TNFα and IL-1β, the anti-fibrogenic factor NF-κβ, the apoptotic factor caspase-3, the proliferative indicator antigen kiel 67 (Ki-67), and the proliferating cell nuclear antigen (PCNA). Furthermore, different studies revealed conflicting findings comparing the therapeutic effects of AD- and BM-MSCs; therefore, another aim of the current study was to critically analyze whether MT-pre-treated AD-MSCs and BM-MSCs reveal equal or different therapeutic and regenerative influences on apoptosis, inflammation, and fibrosis in a CCl4-injured liver experimental rat model.

## 2. Materials and Methods

### 2.1. Isolation, Propagation, and Identification of MSCs

The BM- and AD-MSCs were obtained from the Biochemistry and Molecular Biology Unit, Faculty of Medicine, Cairo University. The femurs and tibiae of 6-week-old male rats were flushed with DMEM (Sigma, St. Louis, MO, USA, D5796) supplemented with ten pecent fetal bovine serum (Sigma, St. Louis, MO, USA, F6178) for the collection of the bone marrow. In a ratio of 2:1, the cells were layered in sterile conical tubes over Ficoll-Hypaque (Sigma, St. Louis, MO, USA, F8016). The cells were then centrifuged. The nucleated cells were isolated and then resuspended in a complete culture medium enhanced with one percent penicillin–streptomycin (Sigma, St. Louis, MO, USA, P4333). The primary culture was performed by incubation of the cells at 37 °C in five percent humidified CO_2_ for fourteen days with complete replacement of the media every four days. At 80% confluence and after the formation of large colonies, the cultures were washed twice with PBS (Sigma, St. Louis, MO, USA, P5493). The cells were then trypsinized with 0.25% trypsin (Sigma, St. Louis, MO, USA, T1426) in 1 mL EDTA (Sigma, St. Louis, MO, USA, E6758) for five minutes at 37 °C and centrifuged at 2400 rpm for 20 min. The first passage was performed with the suspension and incubation of the cell pellets in 25 cm^2^ culture flasks [31,44]. The adipose tissue was taken from the abdomen of Sprague Dawley albino rats after their sacrifice. The digestion of the extracellular matrix was performed with 0.075% type I collagenase (37 °C and five percent CO_2_ for 30 min) and centrifugation at 500 g for five minutes. The pellet was cultured in high-glucose DMEM (Sigma, St. Louis, MO, USA, D5796) with ten percent fetal bovine serum (Sigma, St. Louis, MO, USA, F6178), 2 mML penicillin, streptomycin, and glutamine (Invitrogen, Waltham, MA, USA) and incubated at 37 °C in five percent humidified CO_2_. The experiments were implemented using the AD-MSCs in passage 3 [45]. The identification of the MSCs was determined by their morphology; adherence; detection of the surface markers of the MSCs being positive for CD 34, 45, 90, 105; and their power to differentiate into osteocytes and chondrocytes [46]. The MSCs were labeled with PKH26, a red fluorochrome, according to the manufacturer’s recommendations (Sigma, St. Louis, MO, USA) for in vivo cell tracking. The cell viability was detected by adding a 1:1 ratio of the cell suspension and 0.4% trypan blue stain. The viable cells appeared shiny without staining under a phase contrast microscope.

### 2.2. MT Pre-Treatment of MSCs

For the MT pre-treatment of MSCs, 1 mL of MT (Sigma, St. Louis, MO, USA) was added to 40 mL of ethyl alcohol. The mixture was shaken for two minutes, and then 960 mL of saline was added. The MSCs were exposed to a 24-h pre-conditioning with five μL MT equivalent to 1 mL of the mixture. The mixture three times with PBS (Sigma, St. Louis, MO, USA) for the removal of MT [45,47].

### 2.3. Experimental Animals

A total of sixty adult male Sprague Dawley albino rats, weighing 150–200 g, were used in the present study, and single-blind and stratified randomization processes were applied. The rats were treated following the international guidelines for the care and use of laboratory animals, including the way of animal treatment, anesthesia, methodology of the collection of the MSCs from the animal’s bone marrow, and their use in the experimental research. Minimal animal suffering was ensured. The experiment proposal was approved by the Committee of Research Ethics, Deanship of Scientific Research, Qassim University, approval number 23-20-11. The rats were allowed to acclimatize for two weeks before the experiment, housed in cages under normal light/dark periods, and fed standard food and water ad libitum.

### 2.4. Experimental Design

The experimental rats were divided into the following groups:

Group I (Control, *n* = 10): The rats received subcutaneous injections of 1 mL saline twice weekly for 8 weeks and a single intravenous injection of 0.5 mL PBS into the tail vein.

Group II (CCl4 group, *n* = 10): The rats received subcutaneous injections of 1 mL/kg body weight CCl4 (Sigma, St. Louis, MO, USA) at a ratio of 1:1 with corn oil, twice weekly for 8 weeks, to induce liver fibrosis [48].

Group III (CCl4 and BM-MSCs, *n* = 10): The induction of fibrosis was performed as in group II. The rats received a single intravenous injection of BM-MSCs (1 × 10^6^) diluted in 0.5 mL of PBS into the tail vein at the beginning of week 5 [31].

Group IV (CCl4 and MT-pre-treated BM-MSCs, *n* = 10): The induction of fibrosis was performed as in group II. The rats received a single intravenous injection of MT-pre-treated BM-MSCs (1 × 10^6^) diluted in 0.5 mL of PBS into the tail vein, at the beginning of week 5.

Group V (CCl4 and AD-MSCs, *n* = 10): The induction of fibrosis was performed as in group II. The rats received a single intravenous injection of AD-MSCs (1 × 10^6^) diluted in 0.5 mL of PBS into the tail vein, at the beginning of week 5.

Group VI (CCl4 and MT-pre-treated AD-MSCs, *n* = 10): The induction of fibrosis was performed as in group II. The rats received a single intravenous injection of MT-pre-treated AD-MSCs (1 × 10^6^) diluted in 0.5 mL of PBS into the tail vein at the beginning of week 5.

At the end of week 8, blood samples were collected from the experimental rats by the retro-orbital plexus technique using capillary glass tubes. The collected blood samples were analyzed for the levels of serum alanine aminotransferase (ALT), aspartate aminotransferase (AST), alkaline phosphatase (AKP), and total bilirubin (TBIL). The rats were sacrificed by overdose of intraperitoneal pentobarbital, 40 mg/kg body weight. Then, the liver of each animal was dissected and excised. Half of each liver specimen of each experimental rat was fixed in 10% formaldehyde in PBS at 4 °C and was processed for paraffin blocks and prepared for the light microscopic study. The other half was prepared directly for the gene expression study.

### 2.5. Detection of Studied Genes by Quantitative Real-Time Polymerase Chain Reaction (QRT-PCR)

The samples (0.2 mg) from the liver specimens were homogenized in PBS, pH 7.4, using tissue Lyzer (Qiagen; Hilden, Germany) and then centrifuged at 8000× *g* for 20 min. According to the manufacturer’s protocol, the total RNA was extracted using the RNeasy Mini Kit, cat No. 217004 (Qiagen, Hilden, Germany). The cDNA was synthesized by the reverse transcription reaction using the QuantiTect Reverse Transcription Kit, cat No. 205311 (Qiagen, Hilden, Germany). The gene expression levels for TNF-α, IL-1β, and NF-κβ were amplified from the cDNA using the QuantiTect SYBR Green PCR Kit, cat No. 204141 (Qiagen, Hilden, Germany), and the QuantiTect primer assays, cat No. 249900 (Rn_Tnfrsf1a_1_SG QuantiTect Primer Assay, ID QT00388346; Rn_Il1b_1_SG QuantiTect Primer Assay, ID QT00181657; and Rn_Nfkb2_1_SG, ID QT00396823), with the housekeeping gene ACTB Primer sequence. All the samples were analyzed using the 5-plex Rotor-Gene PCR Analyzer (Qiagen, Hilden, Germany) with the 2ΔΔCt method [44,49].

### 2.6. Light Microscopic Study

The liver specimens were fixed and prepared for the histopathological examination. They were stained with hematoxylin and eosin (H & E) stain to study the alterations in the histological architecture, Masson’s trichrome stain to determine the collagen fibers, and periodic acid–Schiff (PAS) stain to demonstrate the polysaccharides in the hepatocytes [50,51,52].

### 2.7. Immunofluorescent Study for Ki-67

The liver tissue was fixed in four percent formaldehyde in PBS for 15 min at room temperature and covered with ice-cold 100% methanol, incubated in methanol for 10 min at −20 °C, rinsed in PBS for 5 min, and blocked in a blocking buffer for 60 min. The blocking solution was aspirated, and the anti-Ki-67 Polyclonal Antibody (Invitrogen; ThermoFisher Scientific, Hilden, Germany; Germany Catalogue Number PA5-16785) was applied. The hepatic specimens were incubated in fluorochrome-conjugated secondary antibody diluted in an Antibody Dilution Buffer for two hours at room temperature in the dark (Goat anti-rabbit IgG (H + L) Alexa Fluor 488 Invitrogen; ThermoFisher Scientific, Hilden; Germany Catalogue Number A-11034). The slides were covered with Prolong^®^ Gold Antifade Reagent (#9071) or Prolong^®^ Gold Antifade Reagent with DAPI (#8961). The microscopic examination, using appropriate excitation wavelength, was performed by LABOMED Fluorescence microscope LX400, cat No. 9126000, NY, USA.

### 2.8. Immunohistochemical Reaction

The following primary antibodies were used:

PCNA, which is a cofactor essential for DNA replication, repair, and chromatin remodeling, was detected by the rabbit polyclonal IgG (FL-261; catalog number SC-7907, 200 μg/mL, dilution 1:50, Santa Cruz Biotechnology, Santa Cruz, CA, USA). A positive reaction of the nuclear regeneration is indicated by a brown discoloration of the nuclei in the proliferating cells [53].

The anti-caspase-3 mouse monoclonal primary antibody (Dako Company, Cairo, Egypt, Catalog No. IMG-144A, at a dilution 1/200) was used [54]. The slides were rinsed in PBS, incubated with 2 drops of biotinylated secondary antibody for each section for 20 min, and then rinsed with PBS. The substrate chromogen (DAB) mixture was applied for five min and then rinsed with distilled water. The slides were stained with hematoxylin and then dehydrated and mounted. The cytoplasm of the apoptotic cells being stained brown indicates apoptosis.

### 2.9. Histomorphometric Measurements

The area percent of the collagen fibers, the positive immune reaction for the PCNA, and caspase-3 were measured by an independent observer, using a Leica LAS V3.8 image analyzer computer system (Heerbrugg, Switzerland). Systematic sampling was applied by dividing the specimen into a grid and selecting ten non-overlapping fields at a magnification of 400 at regular intervals in a systematic pattern. The area percent represented the areas of the positive reaction, masked by a binary blue color to the area bounded within a standard measuring frame (7286.783 µm^2^), and the area percent of stained cells was quantified using the Cellsens dimension software ver. 4.2 (Olympus, Tokyo, Japan) [55].

### 2.10. Statistical Analysis

All the measurements were expressed as a mean and standard deviation (±SD) and subjected to statistical analysis using the “SPSS 22” (SPSS, Inc., Chicago, IL, USA) software. Analysis of variance using one-way ANOVA and Tukey’s honest significance difference (HSD) post hoc test was performed for the comparison between the quantitative variables. The results were considered significant when the *p*-value was less than 0.05 [56].

## 3. Results

### 3.1. Biochemical Results

Regarding the four groups treated with MSCs (groups III–VI), ALT was significantly decreased in relation to group II (CCl4-treated group) (*p* = 0.00). ALT was significantly increased in the two groups treated with CCl4 and BM-MSCs (group III) and CCl4 and AD-MSCs (group V) compared with the control and the MT-treated MSC groups (groups IV and VI) (*p* = 0.00). The lowest levels were revealed in groups IV (42.19 ± 3.40) and VI (38.61 ± 1.57), with no significant increase in ALT in group VI (CCl4 and MT-pre-treated AD-MSC group) in relation to the control group (*p* = 1.00) (Table 1, Figure 1).

The four groups treated with MSCs showed a significant decrease in AST in relation to group II (CCl4-treated group) (*p* = 0.00) with no significant difference in relation to the control group and in relation to each other (*p* > 0.05). The four groups treated with MSCs showed significant decreases in AKP and Tbil in relation to group II (CCl4-treated group) (*p* = 0.00) and no significant difference in relation to each other (*p* > 0.05) except in relation to group III (CCl4 and BM-MSCs) (*p* = 0.00) (Table 1, Figure 1).

### 3.2. Real-Time PCR for TNF-α, IL-1β, and NF-κβ Gene Expression

The detection of mRNA demonstrated a significant decrease in the expression of TNF-α in the four groups treated with MSCs (groups III–VI) in relation to group II (CCl4-treated group) (*p* = 0.00) with no significant difference in relation to the control group and between groups III–VI (*p* > 0.05). In addition, the level of IL-1β was significantly decreased in the groups treated with MSCs in relation to group II (CCl4-treated group) (*p* = 0.00), except in group III (*p* = 0.84), and significantly increased in relation to the control group (*p* = 0.00), except for group VI (CCl4 and MT-pre-treated AD-MSC group) (*p* = 0.83). The level of NF-κβ was significantly decreased in the groups treated with MSCs in relation to group II (CCl4-treated group) (*p* = 0.00) and significantly increased in relation to the control group (*p* = 0.00) with a significant decrease in group VI (CCl4 and MT-pre-treated AD-MSC group) in relation to the other three groups treated with MSCs (*p* = 0.00) (Table 2, Figure 2).

### 3.3. Hematoxylin and Eosin Stain

After examination of the control group (group I), the specimens showed normal architecture of the liver tissue around the central vein and the portal triad. The central veins were surrounded by radiated cords of hepatocytes and hepatic sinusoids, with Kupffer cells attached to the endothelium of the sinusoids without any pathological changes. The nuclei appeared vesicular with prominent nucleoli, and some binucleated cells were detected (Figure 3A–C).

The CCl4-treated group (group II) revealed degeneration of the hepatocytes with loss of their normal laminar pattern. Necrosis in the form of nuclear karyolysis and karyorrhexis was seen with the dissolution of the cytoplasm and vacuolar degeneration of the hepatocytes. Inflammatory cellular infiltration, sinusoidal congestion, dilated and congested portal vein with portal edema, periportal fibroplasia, and hemosiderosis with bile duct dilatation and hyperplasia were observed (Figure 3D–F and Figure 4A).

Group III (CCl4- and BM-MSC-treated group) showed moderate vacuolar degeneration of the hepatocytes and bile duct dilatation and hyperplasia with minimal inflammatory cellular infiltration (Figure 4B,C). Group IV (CCl4 and MT-pre-treated BM-MSC group) showed normal hepatocytes with minimal periportal inflammatory cellular infiltration (Figure 4D). Group V (CCl4- and AD-MSC-treated group) showed minimal bile duct dilatation and hyperplasia, minimal inflammatory cellular infiltration, and periportal leukocytic infiltration (Figure 4E,F). Group VI (CCl4 and MT-pre-treated AD-MSC group) showed normal hepatocytes and Kupffer cells (Figure 4G,H).

### 3.4. Masson’s Trichrome Stain

The specimens of the control group (group I) showed a minimal amount of collagen fibers around the periportal area (Figure 5A,B). Group II (CCl4-treated group) revealed an increased amount of collagen fibers around the periportal area (Figure 5C,D). The specimens of group III (CCl4- and BM-MSC-treated group) showed a moderate amount of collagen fibers around the periportal area (Figure 5E,F). Group IV (CCl4 and MT-pre-treated BM-MSC group) showed a minimal amount of collagen fibers around the periportal area (Figure 6A,B). Group V (CCl4- and AD-MSC-treated group) showed a moderate amount of collagen fibers around the periportal area (Figure 6C,D). Group VI (CCl4 and MT-pre-treated AD-MSC group) showed a minimal amount of collagen fibers around the periportal area (Figure 6E,F).

### 3.5. PAS Stain

The specimens of the control group (group I) showed normal storage of glycogen in the healthy hepatocytes and a strong reaction of the PAS stain (Figure 7A). Group II (CCl4-treated group) showed a marked decrease in PAS-stained hepatic lobules (Figure 7B). Group III (CCl4- and BM-MSC-treated group) showed a decrease in PAS-stained hepatic lobules (Figure 7C). Groups IV (CCl4 and MT-pre-treated BM-MSC group) and V (CCl4- and AD-MSC-treated group) showed an increase in PAS-stained hepatic lobules (Figure 7D,E). Group VI (CCl4 and MT-pre-treated AD-MSC group) showed normal PAS-stained hepatic lobules (Figure 7F).

### 3.6. Immunofluorescent Study for Ki-67

The specimens of the control group showed a moderate immunofluorescent yellow-green reaction to the marker of proliferation Ki-67 (Figure 8A), while the CCl4-treated group (group II) showed a minimal yellow-green reaction to Ki-67 (Figure 8B). The specimens of the CCl4- and BM-MSC-treated group and CCl4 and MT-pre-treated BM-MSC group (groups III and IV) revealed a moderate immunofluorescent yellow-green reaction to Ki-67 (Figure 8C,D). The specimens of the CCl4- and AD-MSC-treated group and CCl4 and MT-pre-treated AD-MSC group (groups V and VI) revealed an increased immunofluorescent yellow-green reaction to Ki-67 (Figure 8E,F).

### 3.7. Immunohistochemical Reaction

The liver specimens of the rats of group I (control group) showed few PCNA reactions in the form of brown nuclei of the hepatocytes. Group II (CCl4-treated group) revealed almost negative PCNA reactions. The liver specimens of group III (CCl4- and BM-MSC-treated group) revealed a moderate number of brown nuclei of the hepatocytes. Groups IV and V (CCl4 and MT-pre-treated BM-MSC group and CCl4- and AD-MSC-treated group) showed an increased number of brown nuclei of the hepatocytes. Almost all the nuclei of the hepatocytes showed brown coloration in group VI (CCl4 and MT-pre-treated AD-MSC group) (Figure 9A–F).

Regarding the immune reaction to caspase-3, the hepatocytes of groups I, IV, V, and VI showed an absence of brown discoloration of the hepatocytes indicating the absence of caspase-3 reaction, apoptosis. Group III showed a minimal brown discoloration of the hepatocytes, while group II revealed a massive brown discoloration (Figure 10A–F).

### 3.8. Histomorphometric Results

The mean area percent of the collagen fibers was significantly decreased in the four groups treated with MSCs, in relation to group II (CCl4-treated group) (*p* = 0.00), while there was no significant difference between the groups treated with MT-treated MSCs (groups IV and VI) and the control group (*p* = 0.13) and (*p* = 0.91). No statistical difference was observed in the analysis of the PAS area percent in group III (CCl4 and BM-MSCs) in relation to group II (CCl4-treated group) (*p* = 0.89), with a significant decrease in relation to all the other groups (*p* = 0.00). Group VI (CCl4 and MT-pre-treated AD-MSC group) revealed higher area percent for the PAS reaction (18.63 ± 1.34), where there was no significant difference in relation to the control group, and there was a significant increase in relation to all the other groups (*p* = 0.00). The area percent of the PCNA reactions in the four groups treated with MSCs was significantly increased in relation to group II (CCl4-treated group), the control group, and each other (*p* = 0.00). A higher PCNA reaction was revealed in group VI (CCl4 and MT-pre-treated AD-MSC group) (83.91 ± 6.31). The mean area percent of the immune reaction of caspase-3 in the four groups treated with MSCs was significantly decreased in relation to group II (CCl4-treated group) (*p* = 0.00) with no significant difference in relation to the control group and between the four groups treated with MSCs (*p* > 0.05) (Table 3, Figure 11).

## 4. Discussion

Classically, when evaluating the liver function, the procedures will include the measurement of the levels of ALT, AST, AKP, and TBIL as “liver function tests”. However, this term is a misnomer as many of the elements measured in these tests do not indicate the function of the liver but rather determine the source of the damage. According to Vagvala and O’Connor [57] and Kwo et al. [58], a hepatocellular injury could be denoted by ALT and AST elevations disproportional to the levels of AKP and TBIL. However, an elevation of AKP and TBIL out of proportion to ALT and AST indicates obstructive or cholestasis causes. The elevation of all these parameters represents a mixed injury, which was determined in the findings of the present study. These results indicate hepatocellular injury which is denoted by the histopathological alterations in the hepatocytes together with the bile duct dilatation, indicating cholestasis and obstructive jaundice in the CCl4 group, in agreement with Yacout et al. [59]. The authors detected the elevation of these serum marker enzymes in rats receiving CCl4 due to liver damage from this hepatotoxic agent. The moderate changes in the groups receiving both BM- and AD-MSCs indicate the potential therapeutic effects of MSCs, especially in group VI (CCl4 and MT-pre-treated AD-MSCs), which appeared almost normal. The levels of ALT, AST, AKP, and TBIL in only group VI had no significant difference compared with the control group, indicating that AD-MSCs represent a greater potential for regenerative effects than BM-MSCs. Moreover, the MT pre-treatment increased their efficacy compared with group V, in which the AD-MSCs were not pre-treated with MT. In this context, Yilmaz and Karakayali [60] reported that AD-MSCs revealed a greater improvement in liver function than BM-MSCs in the CCl4 liver injury model. The authors described the main mechanism of improving liver function tests as the ability of AD-MSCs to inhibit hepatocyte degeneration and necrosis with enhancement of their survival rates together with their anti-inflammatory and anti-apoptotic actions. Concerning MSC migration and differentiation, it has been revealed that MSCs are attracted to the liver by multiple chemotactic signals such as the concentration gradient of stromal cell-derived factor-1 (SDF-1) present in the liver sinusoid endothelial cells. SDF-1 enhances the activation of α4β1 integrin of MSCs for firm adherence to endothelial cells [61]. The overexpression of the chemokine receptor (CXCR7), under hypoxic conditions, stimulates the migration of MSCs by increasing the levels of vascular cell adhesion molecule-1 (VCAM-1) and CD 44 [33]. The hepatic specification of MSCs is induced by fibroblast growth factor 2 (FGF2), fibroblast growth factor 4 (FGF4), and hepatocyte growth factor (HGF) [62]. The hepatogenesis of MSCs in the injured liver is triggered through the integrin pathway and the downregulation of focal adhesion kinase (FAK) and integrin-linked kinase (ILK) leading to a significant increase in hepatogenic molecules such as HGF, oncostatin M (OSM), and bFGF in MSCs [63]. MSCs were shown to affect the gene expression of TNF-α, IL-6, HGF, and TGF-β, which influence hepatocyte proliferation [44]. The immunomodulation capacity of MSCs is attributed to the intercellular contact and the secretion of the signaling factors acting on various immune cells. MSCs were shown to inhibit the activation of the hepatic stellate cells through the prevention of the proliferation of CD8+ cytotoxic T lymphocytes and CD4+T-helper (Th) cells and the stimulation of the regulatory T cells through the secretion of TGF-β, prostaglandin E2 (PGE2), and human leukocyte antigen-5 (HLA-G5) [64,65]. MSCs eliminate the profibrotic environment under the effect of IL-4 and IL-10. The phosphorylation of anti-apoptotic AKT produces an overexpression of IL-10 and TGF-β and reduces the secretion of IL-12 which enhances the differentiation of mature type 1 dendritic cells (DC1s) to the tolerogenic type 2 dendritic cells (DC2s) in the presence of IL-6 and HGF [66,67]. MSCs affect the polarization of macrophages, decreasing the gene expression of the pro-inflammatory mediator TGF-β1, which is considered the chief molecule in the activation of hepatic stellate cells [68], and increasing the secretion of the milk fat globule-EGF factor 8 (MFGE8), which reduces the TGF-β type I receptor (TGFBR1) [69]. Other pro-inflammatory cytokines, such as interferon-γ (IFN-γ), IL-6, IL-1β, IL-12, IL-4, and TNF-α, were reduced by the secretion of PGE2, TNF-α stimulated gene/protein 6 (TSG-6), granulocyte-colony stimulating factor (G-CSF), and IL-6. On the other hand, MSCs stimulate the production of anti-inflammatory mediators IL-1 and IL-4 and hepatic nuclear factor-4 alpha (HNF-4α) by increasing nitric oxide synthase (iNOS) through the activation of the NF-κB pathway [70,71]. Another effect of the NF-κB pathway is the modulation of hepatic stellate cell proliferation through mediating reactive oxygen species [17]. As shown in the results of the present study, TNF-α and IL-1β were decreased and NF-κB was elevated in the groups treated with MSCs, with the highest expression in the MT-pre-treated AD-MSCs. These findings could overcome the limitations of harvesting BM-MSCs such as the limited number obtained, the signs of senescence during expansion, and the pain and morbidity as a result of the bone marrow aspiration process as reported by Dmitrieva et al. [39]. On the contrary, AD-MSCs could be obtained from multiple sites with minimal invasion and abundant numbers [72,73]. Hao et al. [74] concluded that AD-MSCs significantly reduced the proliferation and activation of hepatic stellate cells more than BM-MSCs. In addition, TGF-B1 was secreted in the culture medium of AD-MSCs at significantly higher levels than those of BM-MSCs. However, the levels of IL-10 and VEGF did not differ significantly between these MSC types. Moreover, the authors denoted that AD-MSCs promoted anti-inflammatory and anti-fibrotic effects on liver tissue more than BM-MSCs, but still, these results were not significant. These findings are in agreement with Rengasamy et al. [75], yet this improvement did not reach the statistical threshold of a significant difference in their findings. Although the present study revealed the same results, they were statistically significant. However, Al-Dhamin et al. [76] found similar effects of both BM-MSCs and AD-MSCs attenuating liver fibrosis. They added that these cells inhibit the activation and proliferation of hepatocyte stellate cells and reduce their apoptosis by the same mechanisms. In addition, the current work examined the proliferative activity of AD- and the BM-MSCs using the immunofluorescence and the immunohistochemistry staining of the proliferative indicators Ki-67 and PCNA, respectively, which revealed increased expression in the group receiving AD-MSCs, which were more effective in the MT pre-treated group.

MT was demonstrated to produce an antioxidant influence protecting the hepatocytes from free radical injury and inhibiting pro-inflammatory cytokines such as TNF-α and IL-1β, retarding the development of liver fibrosis [77]. MT intensely inhibits the neutrophils, mononuclear cells, macrophages, and mast cells which decrease hepatocyte degeneration and necrosis and produce a reduction in the deposition of the extracellular matrix [78]. Czaja [79] reported that MT reduced the serum transaminase activity and decreased the hepatic fibrosis scores as a result of the high expression of NF-κB in the liver tissue. These findings are in agreement with those of the current study. The groups of MT-pre-treated BM- and AD-MSCs revealed improved biochemical, histopathological, and gene expression parameters, with normal levels in the AD-MSC-treated group. MT was found to attenuate the expression of the apoptotic factor Bax in CCl4-induced hepatic fibrosis [80]. Pro-inflammatory factors activate the anti-apoptotic hepatic protein kinase (AKT) phosphorylation that affects the apoptotic gene expression levels of Bax and caspase-3. AKT was proven to play an important role in hepatic stellate cell activation and collagen synthesis, stimulating extracellular matrix deposition and promoting liver fibrosis [13,81]. Moreover, MT was found to promote the functionality of MSCs, improving their therapeutic potential [82]. In addition, MT improved the biological activities and the proliferation of the AD-MSCs, in agreement with the results of the current study. These findings were confirmed in the current work as TNF-α, IL-1β, and caspase-3 reactions were reduced in the experimental groups receiving MSCs with the addition that the MT-pre-treated AD- and BM-MSCs revealed normal parameters. Additionally, MT was determined to enhance the self-renewal potential of MSCs, improve their functional activity [42,47], and inhibit their replicative senescence, especially that of AD-MSCs, acting as an anti-aging agent [43]. In agreement with the current work, the pro-inflammatory cytokines TNF-α and IL-1β, known to induce cellular senescence [83], were reduced in the groups that received MT-treated MSCs, with the lowest levels in the AD-MSC group. Several limitations were raised irrespective of the therapeutic potential of the MSCs. They include the prime use of autologous adult stem cells, the in vitro expansion in compliance with the laboratory procedures of the manufacturer, and the limited number of the cells produced that constrains their mass use. Further investigations are recommended to study the potential therapeutic effects of MSCs on the advanced phases of liver cirrhosis as most of the cases are discovered at this stage. Therefore, potential future studies that include extended follow-up periods are suggested to explore the durability of the observed findings.

## 5. Conclusions

It was observed that both the BM-MSCs and the AD-MSCs improved the functions of the liver cells and reduced the histopathological alterations in the liver tissue. The optimal findings that revealed normal parameters were found in the AD-MSC group pre-treated with MT. In addition, the immunomodulatory influence of this group showed the best results expressed in the levels of the anti-inflammatory, anti-fibrotic, apoptotic, and proliferative factors: TNF-α, IL-1β, NF-κB, caspase-3, Ki67, and PCNA. Therefore, it can be concluded that AD-MSCs pre-treated with MT represent a very significant tool for future stem cell regenerative and therapeutic strategies for the treatment of liver diseases.

## Figures and Tables

**Figure 1 biomolecules-14-00297-f001:**
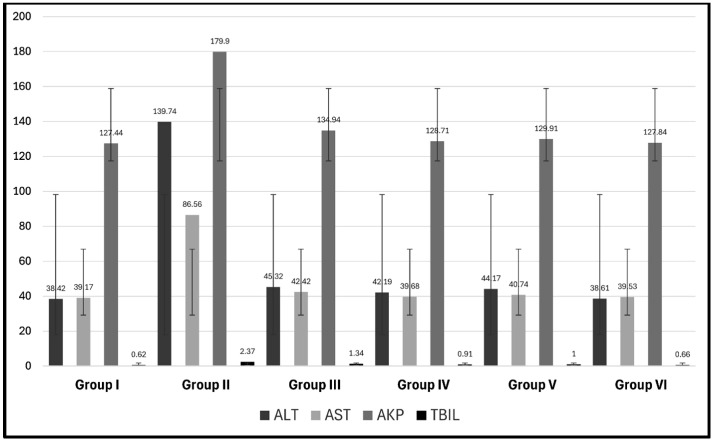
Mean values ± standard deviation of the levels of serum enzymes.

**Figure 2 biomolecules-14-00297-f002:**
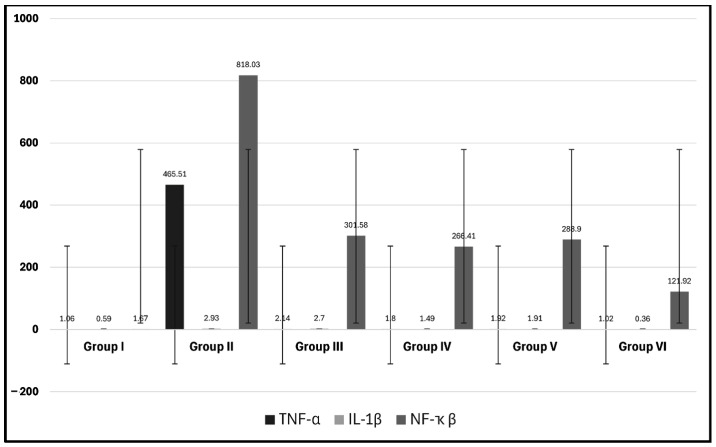
Mean values ± standard deviation of gene expression of TNF-α, IL-1β, and NF-κβ.

**Figure 3 biomolecules-14-00297-f003:**
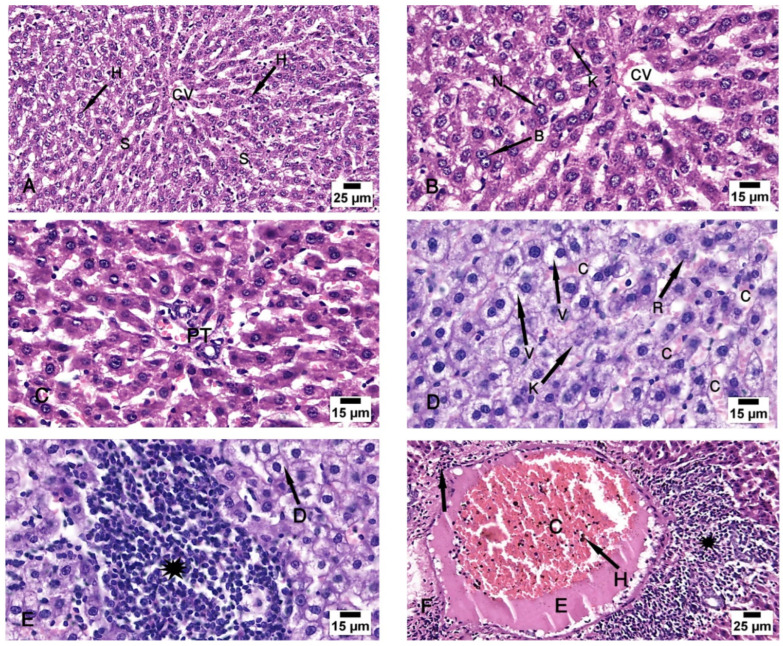
Photomicrographs of the liver tissue. (**A**–**C**) Group I shows the normal architecture of the liver tissue around the central vein (CV) and the portal triad (PT). The central vein is surrounded by radiated cords of hepatocytes (H) and hepatic sinusoids (S). The Kupffer cells (K) are seen attached to the endothelium of the sinusoids. The nuclei appeared vesicular with prominent nucleoli (N), and some binucleated cells are noted (B). (**D**–**F**) Group II shows degeneration of the hepatocytes with loss of their normal laminar pattern. Necrosis in the form of nuclear karyolysis (K) and karyorrhexis (R) with sinusoidal congestion (C) and vacuolar degeneration of the hepatocytes (V) are seen. Inflammatory cellular infiltration (asterisk), dissolution of the cytoplasm (D), dilated and congested (C) portal vein with portal edema (E), periportal fibroplasia (arrow), and hemosiderosis (H) are observed. (H & E; (**A**,**F**) ×200; (**B**–**E**) ×400).

**Figure 4 biomolecules-14-00297-f004:**
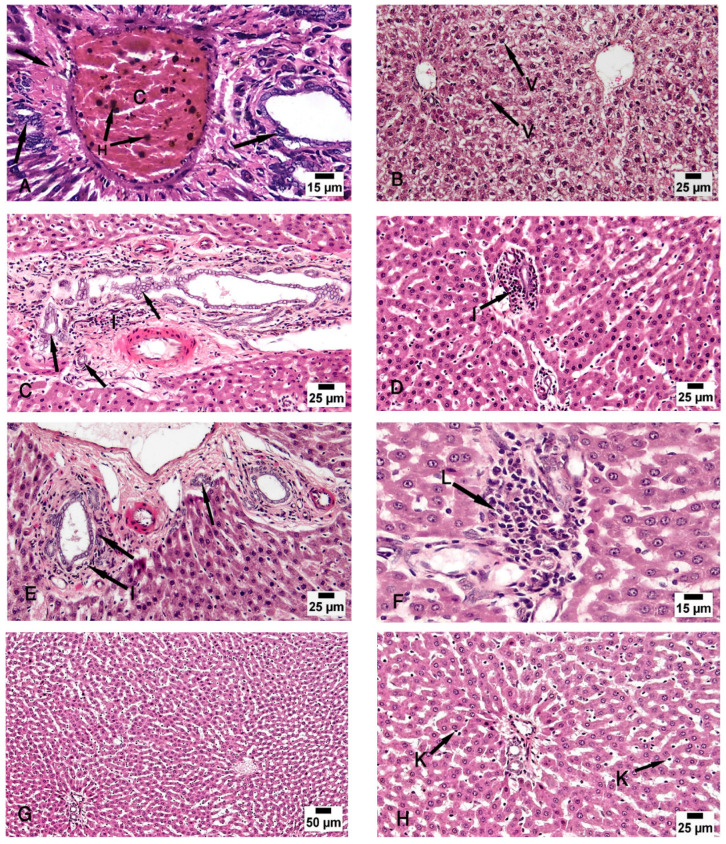
Photomicrographs of the liver tissue. (**A**) Group II shows periportal fibroplasia, bile duct dilatation and hyperplasia (arrow), and congested (C) portal vein with hemosiderosis (H). (**B**,**C**) Group III shows moderate vacuolar degeneration of the hepatocytes (V) and bile duct dilatation and hyperplasia (arrow) with minimal inflammatory cellular infiltration (I). (**D**) Group IV shows normal hepatocytes with minimal periportal inflammatory cellular infiltration (I). (**E**,**F**) Group V shows minimal bile duct dilatation and hyperplasia (arrow), minimal inflammatory cellular infiltration (I), and periportal leukocytic infiltration (L). (**G**,**H**) Group VI shows normal hepatocytes and Kupffer cells (K). (H & E; (**A**,**F**) ×200; (**B**–**E**,**H**) ×400; (**G**) ×100).

**Figure 5 biomolecules-14-00297-f005:**
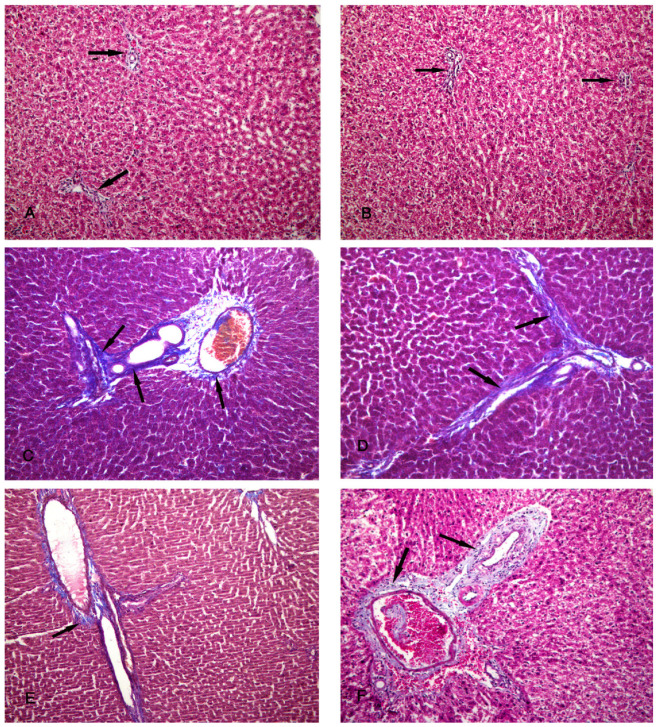
Photomicrographs of the liver tissue. (**A**,**B**) Group I shows a minimal amount of collagen fibers around the periportal area (arrows). (**C**,**D**) Group II shows an increased amount of collagen fibers around the periportal area (arrows). (**E**,**F**) Group III shows a moderate amount of collagen fibers around the periportal area (arrows). Masson’s trichrome ×200.

**Figure 6 biomolecules-14-00297-f006:**
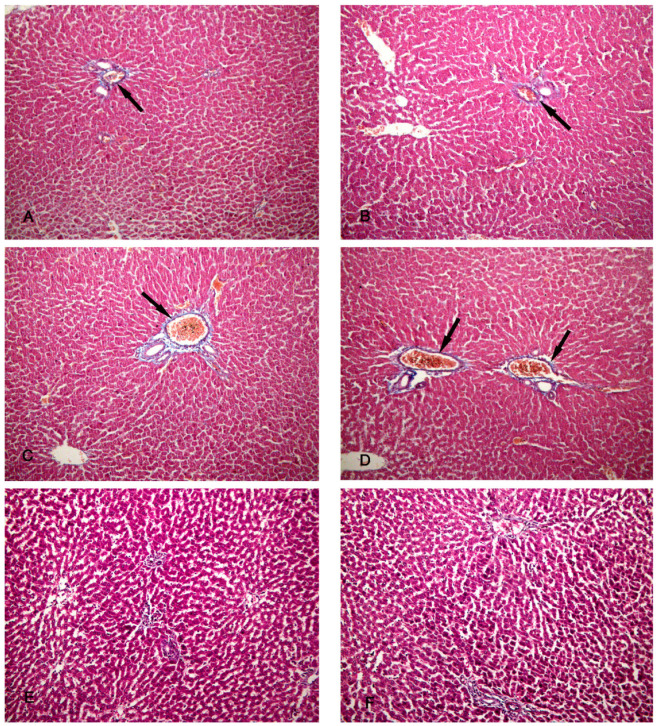
Photomicrographs of the liver tissue. (**A**,**B**) Group IV shows a minimal amount of collagen fibers around the periportal area (arrows). (**C**,**D**) Group V shows a moderate amount of collagen fibers around the periportal area (arrows). (**E**,**F**) Group VI shows a minimal amount of collagen fibers around the periportal area (arrows). Masson’s trichrome ×200.

**Figure 7 biomolecules-14-00297-f007:**
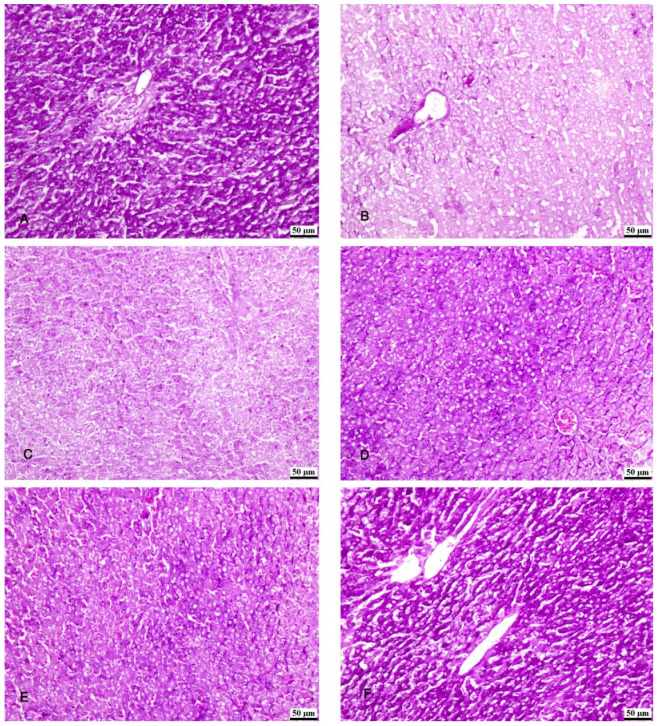
Photomicrographs of the liver tissue. (**A**) Group IV shows normal PAS-stained hepatic parenchyma. (**B**) Group II shows a marked decrease in PAS-stained hepatic lobules. (**C**) Group III shows a decrease in PAS-stained hepatic lobules. (**D**,**E**) Groups IV and V show an increase in PAS-stained hepatic lobules. (**F**) Group VI shows normal PAS-stained hepatic lobules. PAS ×200.

**Figure 8 biomolecules-14-00297-f008:**
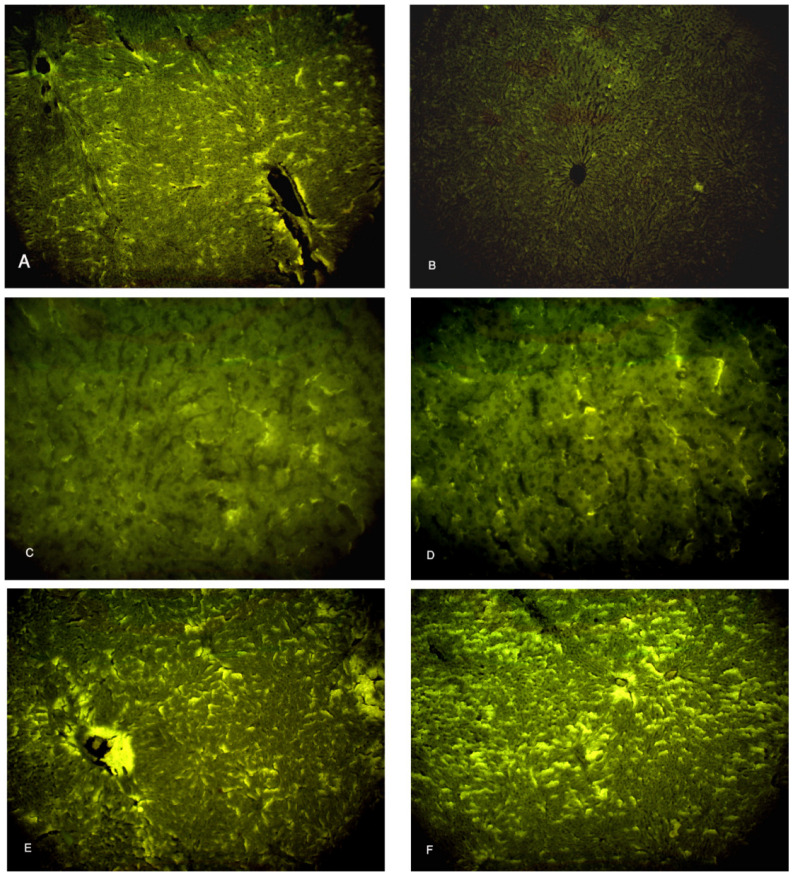
Photomicrographs of the liver tissue. (**A**) Group I shows a moderate immunofluorescent yellow-green reaction. (**B**) Group II shows a minimal immunofluorescent yellow-green reaction. (**C**,**D**) Groups III and IV show moderate immunofluorescent yellow-green reactions. (**E**,**F**) Groups V and VI show increased immunofluorescent yellow-green reactions. Ki-67 ×100.

**Figure 9 biomolecules-14-00297-f009:**
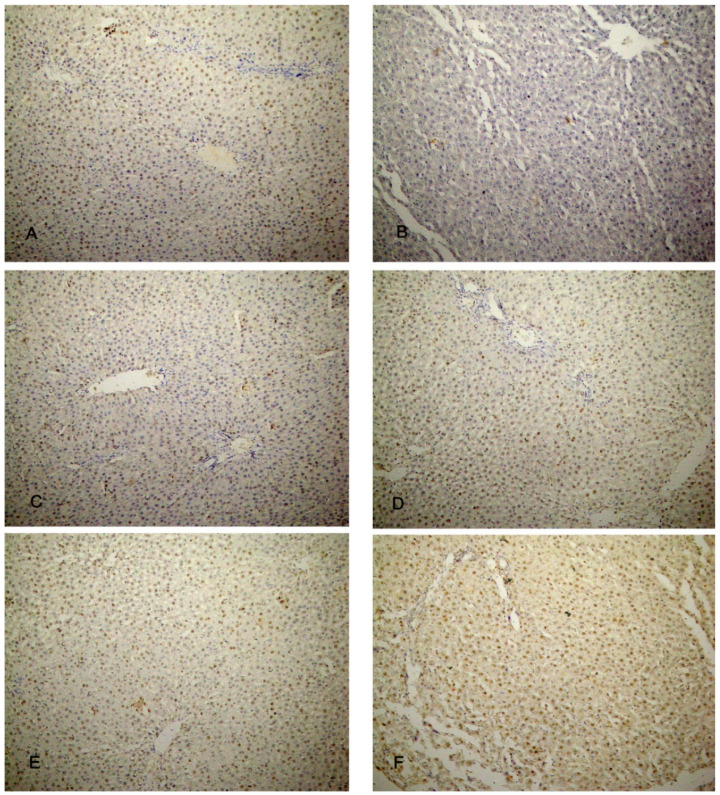
Photomicrographs of the liver tissue. (**A**) Group I shows few brown nuclei of the hepatocytes. (**B**) Group II shows an almost negative PCNA reaction in the nuclei. (**C**) Group III shows a moderate number of brown nuclei of the hepatocytes. (**D**,**E**) Groups IV and V show an increased number of brown nuclei of the hepatocytes. (**F**) In Group VI, almost all the nuclei show brown coloration (PCNA ×200).

**Figure 10 biomolecules-14-00297-f010:**
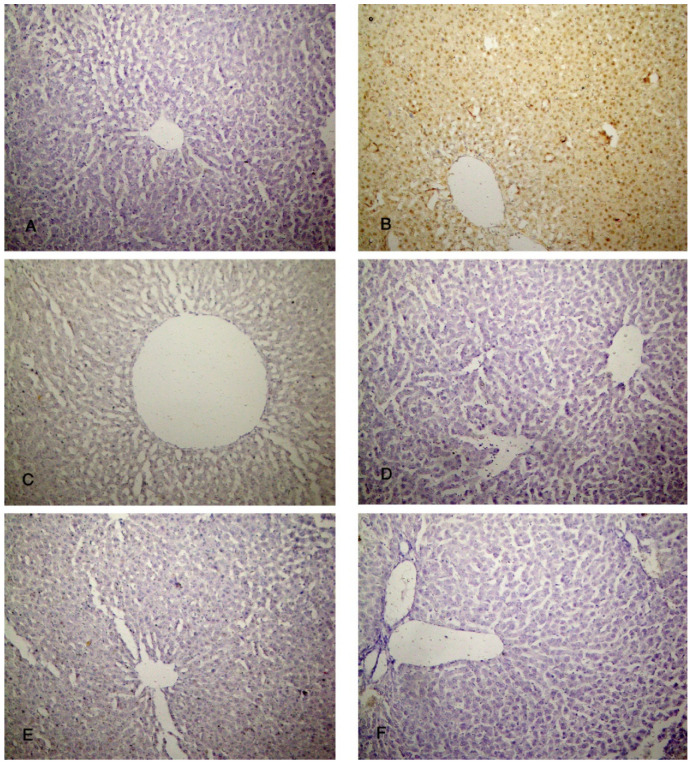
Photomicrographs of the liver tissue. (**A**) Group I, (**D**) group IV, (**E**) group V, and (**F**) group VI show an absence of brown discoloration of the hepatocytes. (**B**) Group II shows a massive brown discoloration of the hepatocytes. (**C**) Group III shows a minimal brown discoloration of the hepatocytes. Caspase-3 ×200.

**Figure 11 biomolecules-14-00297-f011:**
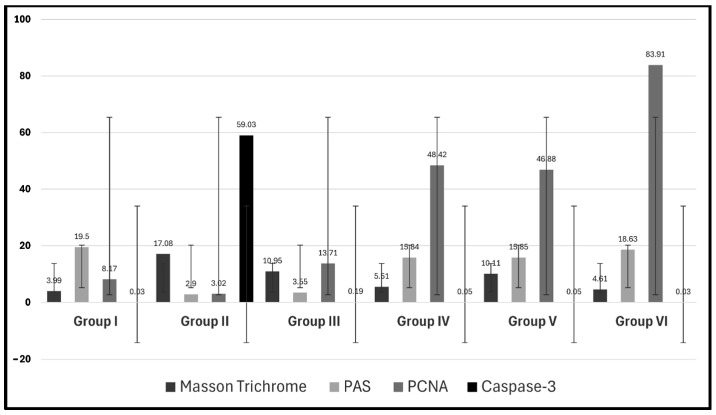
Mean values ± standard deviation of the area percent of Masson trichrome, PAS stains, PCNA, and caspase-3 reactions.

**Table 1 biomolecules-14-00297-t001:** Mean values ± standard deviation of the levels of serum enzymes.

Groups	ALT (U/mL)	AST (U/mL)	AKP (U/mL)	TBIL (mg/dL)
I	38.42 ± 1.08	39.17 ± 1.50	127.44 ± 3.44	0.62 ± 0.24
II	139.74 ± 4.50	86.56 ± 7.53	179.93 ± 5.53	2.37 ± 0.37
III	45.32 ± 2.38 ^a,b,f^	42.42 ± 1.28 ^b^	134.94 ± 4.78 ^a,b,d,e,f^	1.34 ± 0.36 ^a,b,d,e,f^
IV	42.19 ± 3.40 ^a,b^	39.68 ± 1.35 ^b^	128.71 ± 1.43 ^b,c^	0.91 ± 0.17 ^b,c^
V	44.17 ± 2.82 ^a,b,f^	40.74 ± 0.91 ^b^	129.91 ± 1.88 ^b,c^	1.00 ± 0.22 ^b,c^
VI	38.61 ± 1.57 ^b,c,e^	39.53 ± 0.84 ^b^	127.84 ± 3.38 ^b,c^	0.66 ± 0.18 ^b,c^

^a^ compared with group I, ^b^ statistically significant compared with group II, ^c^ statistically significant compared with group III, ^d^ statistically significant compared with group IV, ^e^ statistically significant compared with group V, ^f^ statistically significant compared with group VI.

**Table 2 biomolecules-14-00297-t002:** Mean values ± standard deviation of gene expression of TNF-α, IL-1β, and NF-κβ.

Groups	TNF-α	IL-1β	NF-κβ
I	1.06 ± 0.34	0.59 ± 0.24	1.67 ± 0.23
II	465.51 ± 82.54	2.93 ± 0.44	818.03 ± 28.08
III	2.14 ± 0.27 ^b^	2.70 ± 0.40 ^a,d,e,f^	301.58 ± 30.09 ^a,b,f^
IV	1.80 ± 0.14 ^b^	1.49 ± 0.58 ^a,b,c,f^	266.41 ± 45.61 ^a,b,f^
V	1.92 ± 0.05 ^b^	1.91 ± 0.54 ^a,b,c,f^	288.90 ± 35.20 ^a,b,f^
VI	1.02 ± 0.48 ^b^	0.36 ± 0.19 ^b,c,d,e^	121.92 ± 15.59 ^a,b,c,d,e^

^a^ statistically significant compared with group I, ^b^ statistically significant compared with group II, ^c^ statistically significant compared with group III, ^d^ statistically significant compared with group IV, ^e^ statistically significant compared with group V, ^f^ statistically significant compared with group VI.

**Table 3 biomolecules-14-00297-t003:** Mean values ± standard deviation of the area percent of Masson trichrome, PAS stains, PCNA, and caspase-3 reactions.

Groups	Masson Trichrome	PAS	PCNA	Caspase-3
I	3.99 ± 0.55	19.50 ± 1.78	8.17 ± 0.75	0.03 ± 0.02
II	17.08 ± 1.89	2.90 ± 0.95	3.02 ± 0.72	59.03 ± 4.28
III	10.95 ± 1.93 ^a,b,d,f^	3.55 ± 0.91 ^a,d,e,f^	13.71 ± 3.27 ^a,b,d,e,f^	0.19 ± 0.25 ^b^
IV	5.51 ± 0.56 ^b,c,e^	15.84 ± 1.45 ^a,b,c,f^	48.42 ± 4.44 ^a,b,c,f^	0.05 ± 0.04 ^b^
V	10.11 ± 1.53 ^a,b,d,f^	15.85 ± 1.53 ^a,b,c,f^	46.88 ± 4.53 ^a,b,c,f^	0.05 ± 0.04 ^b^
VI	4.61 ± 0.72 ^b,c,e^	18.63 ± 1.34 ^b,c,d,e^	83.91 ± 6.31 ^a,b,c,d,e^	0.03 ± 0.02 ^b^

^a^ statistically significant compared with group I, ^b^ statistically significant compared with group II, ^c^ statistically significant compared with group III, ^d^ statistically significant compared with group IV, ^e^ statistically significant compared with group V, ^f^ statistically significant compared with group VI.

## Data Availability

Data are contained within the article.

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
