# Peer review of "Comparison between the Regenerative and Therapeutic Impacts of Bone Marrow Mesenchymal Stem Cells and Adipose Mesenchymal Stem Cells Pre-Treated with Melatonin on Liver Fibrosis"

_biomolecules, 2024, doi:10.3390/biom14030297_

Round 1
Reviewer 1 Report
Comments and Suggestions for Authors
This report aims to comprehensively compare the immunomodulatory, anti-inflammatory, and anti-apoptotic effects of MT pre-treated adipose tissue-derived (AD) and bone marrow-derived (BM) MSCs. Additionally, it critically analyzes whether MT-treated AD-MSCs and BM-MSCs exhibit equal or distinct therapeutic influences on apoptosis, inflammation, and fibrosis in CCl4-injured rat livers.
Minor revisions
Introduction
1. While the introduction provides a comprehensive overview of liver cirrhosis and fibrosis, it lacks specific mention of recent groundbreaking studies or advancements in understanding the molecular mechanisms, potentially limiting the currency and depth of the review.
2. The introduction introduces the therapeutic potential of mesenchymal stem cells (MSCs) without addressing potential challenges, controversies, or limitations associated with MSC-based therapies, which could provide a more balanced and nuanced perspective on their application in liver fibrosis treatment. Please add relevant information and include PMID: 25132856 and PMID: 22167649
3. Lack of Detailed Methodology Description: The methodology lacks detailed information on the blinding process or randomization procedures for allocating animals to different experimental groups. These are crucial aspects to ensure unbiased and reliable results, and their omission raises questions about experimental rigor.
4. Limited Reproducibility Information: The isolation and culture of MSCs from bone marrow and adipose tissue are described, but key details such as passage numbers, cell viability assessment, and quality control measures are not explicitly mentioned. Providing these details is essential for the reproducibility of the study.
5. Incomplete Animal Welfare Information: While the number of animals, their weight, and housing conditions are mentioned, the description of efforts to minimize animal suffering or detailed anesthesia and sacrifice methods is lacking. Comprehensive reporting of these aspects is essential for ethical considerations and the scientific validity of the study.
6. Unclear Endpoint Measurement: The histomorphometric measurements section mentions the examination of ten fields per specimen, but it lacks clarity on the criteria used to select these fields. Detailed information on the selection process is crucial for transparency and reproducibility.
7. Statistical Analysis Clarity: The statistical analysis section mentions the use of ANOVA and Post-Hoc tests, but it does not specify which specific Post-Hoc tests were employed. Clear reporting of the statistical methods used enhances the interpretation and credibility of the results.
While the provided results present some positive outcomes of MSCs treatment, there are certain weaknesses and areas of concern in the study. Here are some points to consider:
1. Small Sample Size:
• The text does not provide information on the number of subjects in each group. The statistical significance might be affected if the sample size is too small.
2. Incomplete Statistical Analysis:
• The text mentions statistical significance (P < 0.05) in some instances, but a more detailed statistical analysis, such as the statistical test used, p-values, and correction for multiple comparisons, is not provided. Clear statistical methods and results are essential for the validity of the study.
3. Lack of Control Group for MSCs Alone:
• The study compares groups treated with MSCs and groups treated with both CCl4 and MSCs. However, there is no mention of a control group treated with MSCs alone, making it difficult to assess the specific impact of MSCs.
4. Inconsistencies in Terminology:
• The text uses terminology like "better results," but it lacks clarity on what constitutes a better result. Clear and quantitative measures should be used to communicate the effectiveness of treatments.
5. No Long-term Follow-up:
• The study appears to focus on short-term biochemical and histological changes. A long-term follow-up would be valuable to assess the sustainability and durability of the observed effects.
6. Unclear Methodology Details:
• The text does not provide detailed information about the methods used for MSCs treatment, including dosage, administration route, and timing. These details are crucial for reproducibility.
7. Limited Histological Evaluation:
• While histological evaluations are performed, a more comprehensive set of histological markers could provide a more detailed understanding of liver tissue changes.
8. Incomplete Reporting of Results:
• The text mentions "better results" in groups IV and VI, but specific data points and comparisons are not provided, making it challenging to interpret the significance of these results.
9. No Addressing of Confounding Factors:
• The study does not mention if other factors, such as diet or concurrent medications, were controlled or considered in the analysis.
10. No Conflict of Interest Statement:
• The text does not include a statement about any potential conflicts of interest, which is essential for transparency.
To strengthen the study, it is recommended to address these weaknesses and provide more detailed and transparent reporting of methods and results. Additionally, considering a broader range of assessments and a more extended follow-up period would enhance the robustness of the findings
Weaknesses in Discussion and Conclusion:
1. Lack of Critical Evaluation:
• The discussion lacks critical evaluation of the methodology and limitations of the study. It's essential to discuss potential biases, confounding factors, or limitations that might affect the interpretation of the results.
2. Generalization without Specifics:
• The conclusion makes broad statements about the superiority of AD-MSCs with MT pre-treatment without providing specific quantitative measures or statistical significance. It would be more convincing with detailed statistical outcomes.
3. Inadequate Comparison:
• While the study compares AD-MSCs, BM-MSCs, and MT pre-treatment, there's no thorough discussion comparing these findings with existing literature. A more comprehensive comparison with previous studies would strengthen the conclusion.
4. Incomplete Explanation of Mechanisms:
• The discussion mentions various mechanisms through which MSCs modulate liver regeneration, but it lacks a detailed explanation of how these mechanisms were specifically affected or measured in the study. Providing more clarity on the methodology related to these mechanisms would enhance the discussion.
5. Limited Exploration of Potential Alternatives:
• The conclusion seems to favor AD-MSCs with MT pre-treatment, but it doesn't explore potential alternatives or complementary approaches. Discussing other possible strategies or acknowledging the limitations of the proposed approach would make the conclusion more balanced.
6. No Future Directions:
• The conclusion does not provide insights into potential future directions for research or clinical applications. A forward-looking section discussing the next steps or areas for further investigation would add depth to the conclusion.
7. Unsubstantiated Claims:
• Certain statements lack references or concrete evidence, such as claims about the "best results" or the "very significant tool." It's important to substantiate these assertions with specific data or findings from the study.
8. Overemphasis on Positive Results:
• The conclusion heavily emphasizes positive outcomes, potentially overlooking negative or inconclusive aspects of the study. A more balanced presentation that acknowledges any limitations or challenges would increase the credibility of the conclusion.
9. Ambiguous Language:
• Some sentences, such as "The levels of ALT, AST, AKP, and TBIL in only group VI had no significant difference compared with the control group," are not clear. It would be beneficial to explicitly state whether these differences were statistically significant and to what extent.
10. Repetition of Information:
• Certain points are repeated throughout the discussion and conclusion. Streamlining the content and avoiding unnecessary repetition would make the text more concise and focused.
Author Response
"Please see the attachment."

Reviewer 2 Report
Comments and Suggestions for Authors
The author has designed the study carefully. However I have few comments:
1) The given data supports the author's claim. However, did the author carried out an in vitro study on hepatocytes and demosntrate the efffect of melatoninc treated MSC's on hepatotoxicity.
2) The author hasn't carried out the ICC study for the collagen samples. Is there any particular reason for not carrying out the ICC studies.
3) Did the author characterize the isolated MSCs for the biomarkers specifiic of MSCs?
4) Why did the author chose a concentration of 1 million MSC'sc to transfer intravenously? Did author optimized the concentration of the MSCs for it.
5) Did the author characterizes the secretome and its composition of the MSCs to characterize the effect obtained by the MSCs?
6) Why the author transferrred the MSCs intravenously to study their effect and what was their fate after the transfer?
7) Why diid the author chose cell instead of MSC secretome?
Comments on the Quality of English Language
ok
Author Response
"Please see the attachment."
